# From Reflex to Reflection: Two Tricks AI Could Learn from Us

**Jean-Louis Dessalles**

LTCI, Telecom Paris, Institut Polytechnique de Paris, 91120 Paris, France; jean-louis.dessalles@telecom-paris.fr

**Abstract:** Deep learning and other similar machine learning techniques have a huge advantage over other AI methods: they do function when applied to real-world data, ideally from scratch, without human intervention. However, they have several shortcomings that mere quantitative progress is unlikely to overcome. The paper analyses these shortcomings as resulting from the type of compression achieved by these techniques, which is limited to statistical compression. Two directions for qualitative improvement, inspired by comparison with cognitive processes, are proposed here, in the form of two mechanisms: complexity drop and contrast. These mechanisms are supposed to operate dynamically and not through pre-processing as in neural networks. Their introduction may bring the functioning of AI away from mere reflex and closer to reflection.

**Keywords:** machine learning; complexity; simplicity; cognition; contrast

## 1. Introduction

It is not uncommon among machine learning specialists to consider that virtually any task can be learned, provided an adequate "loss function" is available. This faith in data-driven machine learning has been amply confirmed in recent years, even in domains such as machine translation that require much thinking when performed by humans. A logical extrapolation from this faith is that the power of artificial intelligence is potentially unbounded and will probably lead to the technological singularity, as often prophesized since Good [1] (for a review of such beliefs, see [2]). My purpose here is to mention strong arguments against the possibility that the mere extrapolation of current techniques could bring us any closer even to some form of general intelligence. Comparison with human cognition reveals that qualitative progress is still to be made for artificial intelligence to solve a variety of tasks that are obvious even to small children. Such qualitative progress relies on future discoveries that are impossible to date.

Traditional alternatives to data-driven techniques such as neural networks involve symbolic representations such as predicates, rules or graphs. Though these symbolic techniques may be efficient in certain contexts, e.g., when implemented in ontologies, their limits as a general mechanism to represent intelligence have long been questioned [3,4]. The very idea that "concepts" might correspond to fixed symbolic elements that are combined to form thoughts seems to lead to a variety of paradoxes [5]. Here, we choose to explore an alternative way, which is to consider that intelligent processing might result from a limited repertoire of *mechanisms*. Various fundamental mechanisms have been proposed to serve as basis for general intelligence [6]. Some of them have been invoked more specifically to account for various aspects of cognitive performance, such as the "merge" operation in linguistics, or "abduction" in reasoning. Here, we concentrate on two mechanisms, *complexity drop* and *contrast*, because they illustrate the gap between deep learning techniques and what human beings would expect from intelligent devices, and because they suggest promising ways to bridge that gap.

In what follows, I will first point to various limitations of current machine learning techniques. Many of these shortcomings are regularly pointed out (see [7] for a recent example). However, the list

of points I will raise against current standard machine learning is good to have in mind when reading the rest of the paper. Moreover, some of these points are original or are rarely mentioned. The second and third sections to come are devoted to two mechanisms, complexity drop and contrast, that I am presenting as tentative improvements of deep learning techniques. Though they already have given rise to some implementations, they exist mostly on paper at this stage. I think nevertheless that they are worth mentioning, as it is important not only to show why the techniques that led to the current AI revolution are not omnipotent, but also that they can be qualitatively improved in a way that would bring them closer to human intelligence.

## 2. Some Limitations of Deep Learning

The recent rebirth of artificial intelligence is mainly due to the discovery of new learning techniques for multi-layer neural networks, and to the possibility of implementing them in parallel processors. Thanks to these techniques, it became possible to stack many layers, leading to so-called deep learning networks. Multi-layer networks learn to associate configurations of the input layer with desired configuration of the output layer. A network with more layers has more combinatorial power and can learn from raw data (e.g., images), despite the huge dimensionality of the input space. There are, however, limits to what neural networks can achieve. A few of them are mentioned as follows.

### 2.1. The Continuity Bias

Are deep networks able to learn any association? The answer is no. The general reason is that not all learnable problems can be learned through statistical methods [8]. Deep learning techniques are biased towards solving certain kinds of task. Bias is unavoidable [9] and is necessary for reaching efficiency. It is, however, useful to keep in mind how neural networks are biased. In particular, neural networks implement *continuous* associations. This means that a small variation in the input will produce most of the time a slightly modified output. Such a network can still learn slightly discontinuous functions, i.e., functions that are continuous almost everywhere: the more discontinuities, however, the harder it is to learn the association.

Many decisions can be seen as continuous associations. A bank may want to link customer profiles to probabilities of successful loan repay. The final decision, granting the loan or not, introduces just one discontinuity in an overall continuous function. Many cognitive tasks are, however, discontinuous in nature. This is especially true when reasoning is involved. Criminal investigators would make poor decisions if they based their conclusions on the resemblance between crime situations (age of the victim, age of her children or parents, time of the crime . . . ). Two similar crime scenes may eventually lead to totally different kinds of suspects. Even if statistical typology of crime scenes can be recorded, one cannot base criminal conviction on mere statistics.

### 2.2. Isotropy Bias

Another aspect of the bias inherent in neural networks is rarely mentioned. Neural architectures and learning algorithms are insensitive to various isometric operations that would affect both training and test data. We are speaking here of isometries in the input hypercube space (if there are $N$ neurons in the input layer with a maximum intensity value, the input hypercube has $2^N$ vertices). Isometries in the hypercube consist in adding some constant vector to both training and test data (translations), or in scrambling dimensions (rotation), in changing some coordinates into their opposite (symmetries), or any combination thereof (for binary input, isometries are all combinations of translations and rotations).

The point is that some systems such as traditional neural networks will behave exactly the same if both training and test data are systematically transformed through an isometry. This means that no effect whatsoever will be noticed on the way the network will classify the data [10]. For instance, a traditional neural network would learn to recognize a set of faces equally well if their pixels are shifted in intensity or are scrambled, provided that these shifting or scrambling operations are systematic and concern both training and test data. They would also learn equally well on negative pictures.

This is due to the fact that these networks have no absolute preference for input values or for fixed relations between input values. This property is called *relativity* if it concerns translations, *isotropy* if it concerns rotations, and more generally, *indifference* if it concerns any isometry. The property is generally considered to be an asset, as it confers its generality to the learning method. Any bias that departs from these properties is perceived as "particular", not because it limits what can be learned, since all learning systems are biased [9], but because it introduces some "absolute" limitation. For instance, convolutional networks introduce locality as an absolute bias. They are by design insensitive to non-systematic geometrical translations (an image may or may not be translated and still the recognized). However, since convolution filters detect neighboring relationships, they are sensitive to any systematic permutation of pixels (i.e., rotation in the hypercube) that would destroy neighborhood relations. Convolutional networks are therefore *less* isotropic than traditional neural networks.

This property of being relative, isotropic or indifferent has been introduced to characterize innateness [10]. All learning systems are biased, and therefore part of them is innate. However, biases differ on their indifference to isometries. This indifference measures how much the bias is pre-tuned to some odd properties of what is to be learned. This is why strong indifference is preferred for learners that are thought to be "general purpose".

Relativity and isotropy are not without consequences on what indifferent systems may learn. The classes obtained through indifferent learning are, in some way, "good shapes", which means that they are left globally invariant by many of the isometries that do not affect the learning system. For instance, since most image recognition systems based on neural networks are relative, they tend to detect differences in pixel colors or intensity rather than absolute values of these parameters. So the set of potential images that would be classified as "cat" may accept much luminance variation, even if those variations were not present in data.

Though many recognition tasks benefit from this relativity/isotropy bias, not all learning situations consist in learning good shapes. For instance, due to the central embedding property, phrases in a human language may be included into one another, but may not overlap. Children seem to be biased to expect central embedding from linguistic input [11]. This bias, as several other linguistic properties that Chomsky grants to the child's brain, constitutes a departure from isotropy. Children indeed would have trouble learning from "rotated" sentences (e.g., if the second and third words are systematically permuted), as such operations would destroy phrase structure[1]. When the classes to be learned (here, syntactically correct sentences) cannot be coded in a space in which they have "good shape" properties, we expect neural networks to reach only limited performance, or to require very large data set [7].

### 2.3. Dependence on Large Datasets

One distinctive feature of deep learning is that it requires a huge quantity of data to become operational. By contrast, human children learn new concepts often in one shot, or based on scarce evidence. A child learns about four new words a day [12], plus as many set phrases and nuances, plus lexemes from other languages (including slang). Since the frequency of these words follows a Zipf law, many of them are so infrequent that the child has little chance to hear them twice before puberty. One-shot learning is also possible for faces, music or gestures.

Statistical learning, quite surprisingly, is also able to learn in one shot. Some studies rely on semantic embedding of both data and class labels to guess labels for previously unseen classes, a process sometimes referred to as "zero-shot" learning [13]. It is also possible to learn categories from one single instance in the absence of any labeling. In a study on written character acquisition [14], a system learns

---

[1]　Children would probably "debug" such a syntactically weird language, as they do sometimes by ignoring irregularities, and so they would learn something else than what is presented to them. More extensive systematic permutations would make the language impossible to learn, despite the fact that exactly the same information is presented to the children.

to associate characters from various alphabets together with the sequence of strokes that were used to write them (the dataset reveals that people tend to perform the same strokes in the same order when writing characters, even characters unknown to them). Then, the system is able to match character images by analyzing them as successions of strokes. When presented with a totally new character, the system is able to distinguish subsequent images of the same character (drawn by other hands) from other character images. In other words, once the system has acquired sufficient expertise in the domain (here, once it is able to write, so to say), it can learn in one shot. A similar feat is observed when music lovers can identify a composer when hearing an unknown music, even when they know only one piece of that composer.

These achievements are made possible through the previous acquisition of thorough expertise using large datasets in the domain to which the new learned item belongs. The method reaches its limits when it comes to learning relations and structures.

## 2.4. No Easy Learning of Relations

Neural networks are good at learning static classes, such as telling cats from dogs. Learning relations proves much more difficult. Some studies showed that a deep network can describe the content of pictures using an attention mechanism that isolates salient features in the image. These descriptions may involve relations, as in "a woman is throwing a frisbee in a park" [15]. The system could go beyond what is actually in the scene, the woman and the frisbee, and was able to infer a relation: "throw". It seems that the kind of relations the system is able to extract, based on captions available in the dataset, are relations that are almost systematic (frisbees are systematically pictured when thrown by someone). The problem of relation learning as such has been addressed in other studies [16]. For instance, a network can learn relations such as "is on the right of", "has the same shape as", based on images of simplified scenes. The dataset contains images and associated captions such a "the sphere is on the right of the cube". Thanks to the variability of the arguments across images, the system is able to abstract the relation.

Children, by contrast, learn relations in one shot or in few shots from complex, and often abstract, situations. They learn the meaning of relational words such as "prevent", "around", "chase", "abdicate" early in life, without relying on statistical processing. The cognitive processing underlying one-shot learning of relations remains obscure, and could involve structure detection.

## 2.5. No easy Learning of Structures

IQ tests heavily rely on structure detection. For instance, most people find the task of continuing the sequence 1, 2, 2, 3, 3, 3, 4, 4, 4, 4 obvious. They also comment on their solution by saying something such as "*n* repeated *n* times". This kind of test is way beyond what AI programs can solve (unless they can retrieve the answer from the Web). In particular, it is hard to think of a loss function that would allow a statistical device to solve this kind of riddle. Being blind to structure, a statistical system might propose something as irrelevant as 4, 4, 3, 4 for continuation, just because there are mainly 4s and some 3s in the sample.

The ability to detect structured patterns lies at the core of human intelligence. It has been shown to be crucial in the ability to form analogies. In Hofstadter's problem [17], the point is to solve proportional analogy equations such as:

$$\texttt{abc} \text{ is to } \texttt{abd} \text{ as } \texttt{ppqqrr} \text{ is to } X$$

Most people would answer that "the" solution to this problem is $X = \texttt{ppqqss}$. Making such an answer requires much structural processing: detecting letter increment in abc, matching abc with abd and abstracting the operation 'increment last', detecting letter increment followed by element duplication in ppqqrr, and finally transferring 'increment last' in the last structure, after letter increment but before element duplication. People achieve these operations effortlessly. There have been various attempts to reproduce this ability computationally, but analogy programs remain

complex and do not compare with human performance [18]. Word-embedding techniques have been used to produce analogies among word meanings, such as "`king` is to `queen` as `man` is to `woman`" [19]. Recurrent neural networks can be trained to detect various structures, such as syntactic dependencies [20]. However, neural networks and other statistical techniques are inappropriate to solve structural analogies such as the above, in which the structure to be discovered, is in part new and cannot be inferred from dataset exploitation.

### 2.6. Blind to Exceptions

Neural networks, like any statistical device, are ill-equipped to detect anomalies. If we present an image of a face with one missing eyebrow to a face recognition system, it will be blind to the anomaly, as the missing eyebrow introduces only a small perturbation in comparison with a typical face. Statistical devices can detect anomalies, but only when they know in advance which dimensions are relevant to monitor. Any instance that is several standard deviations away from the mean along those dimensions is then considered as an anomaly [21]. The problem is that in high-dimensional spaces, one does not always know which dimensions are relevant. Standard neural networks offer no mechanism to distinguish between a major departure from the mean along a few dimensions, which would characterize an anomaly, vs. an unremarkable small deviation along many dimensions. This is why a standard neural network would consider that a face with an erased eyebrow is as normal as the same face with a small additional noise.

Many decisions crucially depend on unanticipated idiosyncrasies, some revealing details that make a standard decision inappropriate (think of all defects that would attract your attention when buying a second-hand car). Blindness to exceptions and idiosyncrasies not only affects decision quality, but it also impairs the chances that automated decision systems be accepted and trusted. Failure to detect the singularity of each case has often been mentioned as a critique against decision-making algorithms, especially in law [22].

### 2.7. Negation, Inconsistencies and Explanations

Neural networks are unable to detect not only unusual anomalies, but also impossible ones. For instance, a house that has no door is surprising, not because it is atypical, but because it raises a logical problem: how can people get into it? Since there is no device that connects neural networks to logical reasoning, they cannot detect inconsistencies.

More basically, standard neural networks are unable to negate. For an image recognition system, everything is a cat, more or less. The decision that is eventually taken on top of the network, "this is a dog", does not mean "this is not a cat" or "this is not a car". Because they are in essence continuous devices, neural networks do hesitate. They are most of the time unable to generate clear-cut judgments such as "this is not a cat", which are basic to inconsistency and logical reasoning.

This problem is directly connected to the problem of explanation. Neural networks function as black boxes that generate decisions without being able to explain them [7]. Some attempts to solve this problem consist in building a second device that observes the network in charge of the decision, in order to exhibit the locally most contrastive dimensions that might explain why the decision was negative instead of positive [23]. This strategy might work for binary decisions, as when a system is in charge of granting a loan. It cannot produce explanations such as "this animal is not a cat, because it is too big to be a cat".

### 2.8. Narrow Expertise

The most obvious difference between artificial and human intelligence is that the former is limited to narrow fields of expertise. This is not true of systems, such as IBM's Project Debater[2], that mine existing texts to find arguments in support of any issue. It is true, however, of systems like neural

---

[2] See https://www.ibm.com/events/think/watch/replay/120118800/

networks that build their own expertise. A network trained to play Go [24,25] wouldn't notice anything wrong if the opponent put down a stone of the wrong color, or put down two stones instead of one. The system is so specialized that it knows nothing beyond what is strictly required for its performance, as here the first rule of the game! Statistical AI carries specialization to the extreme: it carries out one specific task; if the input departs qualitatively from the anticipated data, it would produce poor results or no result at all. For instance, a dermatologist deep network trained to recognize melanoma [26] wouldn't detect other skin conditions such as psoriasis or eczema. In contrast to human beings who are able to extend their competence, neural networks only increase their field of expertise at the expense of accuracy, except if some hard-wired devices are introduced to separate tasks [27]. Much hope is placed in transfer learning to extend expertise to situations in which the distribution of data changes, using criteria such as minimal distance or minimal complexity [28,29]. However, these techniques remain limited in scope and their extrapolation is not expected to give rise to general intelligence.

One related issue is that neural networks are unable to use general background knowledge about the world. I remember when I learned the concept "buffet plate clip for wine glass" after seeing only one instance of it. This was only possible because I knew many relevant things: that one needs so to say three hands in stand up buffets to hold the plate, the fork and the glass; that full glasses should be kept vertical; and so on. Again, no extrapolation of current deep learning techniques is expected to make general knowledge available for learning new concepts.

### 2.9. No Sense-Making

Deep learning and word-embedding techniques are now currently used in semantic processing and machine translation. This gives the impression that machines have access to meaning. The kind of mistakes that automated translation makes however reveals that this is not the case [30]. Let us quote Hofstadter:

> "Indeed, what about this freshly coined phrase 'One swallow does not thirst quench' (alluding, of course, to 'One swallow does not a summer make')? I couldn't resist trying it out; here's what Google Translate flipped back at me: 'Une hirondelle n'aspire pas la soif'. This is a grammatical French sentence, but it's pretty hard to fathom. First it names a certain bird ('une hirondelle'–a swallow), then it says this bird is not inhaling or not sucking ('n'aspire pas'), and finally reveals that the neither-inhaled-nor-sucked item is thirst ('la soif'). Clearly Google Translate didn't catch my meaning"

By training a network to learn word-context associations, one can detect semantic proximity among words. However, semantic proximity is not semantics. Meanings extracted from texts are not grounded in perception [31]. This puts severe limits to the kind of intelligence text-based learning can reach. Such system cannot grasp the fact that a dead person is dead forever, and that she does no longer eat or sleep. It cannot understand meanings in context, such as the difference between "behind the rock" and "behind the car" and why the latter is more ambiguous.

### 2.10. No Systematicity

The last example is revealing. The ambiguity is systematically increased when the landmark (here, the car) has a front-rear orientation. For a static observer, "behind the car" may refer to a location between the car and the observer, whereas "behind the rock" refers to a location beyond the rock. Even if some simple relations can be learned from pure statistical evidence [16], and even if a deep network could learn that the meaning of "behind" may be different for rocks and cars, it would have no way to discover the systematicity of the phenomenon. Even when data show systematic symmetry (e.g., faces), a neural net will fail to grasp the property. When in a situation of generating new faces [32], generative adversarial networks generate faces that may violate symmetry requirements.

More generally, it has been suggested that neural networks, by design, are unable to ensure systematic relations [33]. For instance, a trained network may have a representation for `smaller(m,n)` for many couples `(m,n)`, but for other couples the relation would be as meaningless as would be 'air is

smaller than blue' to us [7,34]. The fundamental reason for this lack of systematicity is that statistical learning has only access to the *extension* of relations. The way it generalizes, through interpolation between learned examples, does not allow it to build predicates, and confines it to sets (or, equivalently, to membership functions). A statistical learning device that would learn the extensions of two relations `smaller(`*m*`,`*n*`)` and `larger(`*m*`,`*n*`)` would not be able to see that there is a systematic link (negation) between them.

## 3. Intelligence from an Algorithmic Information Perspective

It is tempting to consider that the limitations listed above could be corrected by more computational power, more storage capacity, more data. This view, pushed to the limit, would consider that a program that fails on an IQ test should be given more examples of IQ tests to become more intelligent. An alternative view consists in finding out new mechanisms that would augment the power of current techniques qualitatively. One such tentative mechanism is linked to the notion of complexity.

Artificial intelligence developed throughout decades mainly as a cumulative set of clever techniques, with only a few theoretical frameworks such as Logic or PAC-learning to organize research efforts. Among available theories, there is one that could play a unifying role: Algorithmic Information Theory (AIT). From its beginnings, the central notion of AIT, known as Kolmogorov complexity, was proposed as a way to solve the problem of induction in machine learning [35]. Various tasks achieved in AI, such as learning, structure detection or conceptual characterization can be analyzed as a reduction of complexity, that is, compression. One of the co-inventors of the notion encapsulated the link between algorithmic information and intelligence in his famous aphorism: "Comprehension is compression" [36]. We can evaluate what statistical learning does and does not from the perspective of AIT. The main point is that neural networks, indeed, do achieve compression, but only a particular form of it.

### 3.1. Intelligence as Compression of Information

AIT measures information as the complexity of objects, i.e., the size of their most concise description. More precisely, the Kolmogorov complexity of an object is the size of the shortest program that can be used to determine that object. As such, the notion can be shown to be uncomputable, as one is never sure to have reached the absolute minimum. However, from an AI perspective, the point is not to determine the absolute information content of an object or a situation, but to approach it. And there are various means to evaluate reasonable upper bounds of Kolmogorov complexity, just by attempting to diminish the quantity of information required to designate the object. Following Chaitin's sentence, intelligent processing occurs whenever a device finds out a description of its input that is shorter than what was available to it previously. Statistical machine learning does achieve compression. After learning, a neural net offers a concise description of the dataset. If the dataset consists of image-class pairs, the system spares $\log_2(n)$ bits for each correctly classified item if there are $n$ different classes, since it predicts the class from the image[3]. Moreover, its generalization capabilities, which result from the continuity of the learned association, ensure that it can compress an unbounded amount of data.

One could argue that image compression algorithms, such as Gif, do achieve compression as well, and yet would not be regarded as clever. The point is that intelligent systems discover how to compress data by themselves. This is certainly the case for neural networks, which converge to the association that produces compression through learning. However, statistical compression is by no means the only form of compression. Another example is structure detection. A system that would discover the structure underlying 1, 2, 2, 3, 3, 3, 4, 4, 4, 4 (without merely retrieving it from the Web) could compress the sequence up to infinity and, because of that, would be regarded as intelligent. Similarly, it has been suggested that the solution of structural analogy problems corresponds to finding

---

[3]　One needs $\log_2(n)$ bits to unambiguously designate one class among $n$.

optimal compression, that is, a minimum of Kolmogorov complexity [37]. The best solution of the proportional analogy "`abc` is to `abd` as `ppqqrr` is to $X$" is the value of $X$ that fits with the shortest available description of the quadruple (`abc`, `abd`, `ppqqrr`, $X$). Applying this principle of structural compression opens the way to new forms of learning that can be applied, for instance to language learning (Murena et al., submitted). For instance, analogy might be involved when transferring knowledge from (talk, talked) to (solve, solved) following a minimal complexity principle, with little or no need for statistics. Note that intelligent behavior is achieved in these examples in the absence of any externally specified loss-function. The system knows that it found a clever solution whenever compression occurs.

Algorithmic information has also been invoked as an optimal way to choose among hypotheses in reinforcement learning [38]: hypotheses consistent with past observations are considered more probable if they have shorter descriptions. In this way, the learner favors futures that correspond to maximal compression (or, equivalently, to minimal surprise).

Compression is also at work in a fundamental aspect of intelligence: anomaly detection. While standard neural networks are unable to detect unanticipated anomalies (e.g., a face with a missing eyebrow), AIT offers a nice characterization of what counts as an anomaly. A "normal" instance requires a lot of information to be distinguished from all other objects or events of the same type. A "normal" face would require a long description to be characterized within a set of faces. By contrast, an "abnormal" instance, such as a face with only one eyebrow, can be characterized concisely by mentioning a feature that makes the event unique or almost unique (with an interesting trade-off between the simplicity of the feature and the number of instances that have it) [39].

The search of minimal Kolmogorov complexity has only recently been acknowledged as a fundamental principle at work in human intelligence. Several authors noticed that human beings are very good at computing the simplicity of situations, and that it could explain various cognitive capabilities [40,41]. We already mentioned that good analogy corresponds to a minimum of complexity. At a lower level, the reconstruction of partially hidden patterns seems to obey a principle of minimum Kolmogorov complexity [40]. Conversely, when human subjects are asked to exhibit random behavior by producing what they regard as unintelligible, structureless sequences, they do so by maximizing the complexity of their responses [42]. The cognitive importance of Kolmogorov complexity, however, is not limited to its role in guiding intelligent processing, as in learning, in analogy making or in structure detection. It is also central to determine what is relevant.

### 3.2. Relevance and Complexity Drop

Relevance is central to intelligence. Attention requires a relevance criterion to determine what the system or organism should focus on. Learning requires a relevance criterion to determine which characteristics should be ignored and what should be memorized. Problem solving requires a relevance criterion to determine which elements may lead to a solution. Action requires a relevance criterion to determine what to do next. And last but not least, verbal interaction requires a relevance criterion to determine what to say. Most artificial intelligence systems use built-in relevance criteria, such as high frequency in learning or novelty for attention. AIT offers a general relevance criterion, which is a further illustration of the role of compression. To be more precise, we need to distinguish between *expected* complexity and *observed* complexity. The central principle of *Simplicity Theory* [39] (see also www.simplicitytheory.science) is that any situation or element of a situation is relevant if it generates a complexity drop between *expected* complexity and *observed* complexity. In other words, relevant situations are simpler than expected. Let's mention a few applications of this principle.

Suppose that today's National Lottery draw turns out to be 1, 2, 3, 4, 5, 6. This draw is much simpler than expected, as its description requires much less information than if six "normal" numbers had to be specified. In AIT's terms, the sequence is compressible. This situation is highly relevant to notice and would trigger a reflex of communication among most people who hear the news. Slightly more complex sequences such as 1, 4, 7, 10, 13, 16 are judged less interesting and more "probable" by

people [43]. In this second example, the mention of the increment, 3, makes the description slightly more complex. The amplitude of the complexity drop gets smaller, and the event is perceived as less relevant. This effect is not predicted by probability theory.

Another illustration of the complexity drop principle is offered by coincidences. On September 10th, 2009, the numbers 4, 15, 23, 24, 35, 42 were drawn by a machine live on the Bulgarian television. The event would have gone unnoticed, were it not that the exact same numbers had come up in the preceding round, a few days before. Again, the event is much simpler than expected, since the draw can be described by mentioning its rank in the list of past rounds, instead of describing all of its numbers. The complexity drop principle predicts that the event would have been slightly less relevant if there had been a three-week interval between the two identical draws, and even less relevant for an interval of three and a half months. It also predicts that an exactly one-year interval would have been more interesting than an interval of three and a half months, just because one year is less complex to describe than three and a half months.

The complexity drop principle can be used to decide which features make a situation relevant. A missing eyebrow can make a face unique, as would the presence of two moles located 9.2 mm and 17.7 mm to the left of the left eye. However, the latter description is complex, as it includes the designation of precise numbers. It may be as complex as the expected complexity, which amounts to $\log_2(N)$ bits if there are $N$ faces in the set, as this is the amount of information needed to discriminate each face. The missing eyebrow description, however, remains simple and may leave us with one unique instance even for large values of $N$. The complexity drop principle does not only provide a criterion to decide what is relevant, but it also says which features are relevant in the description [39]. The presence of moles or the color of the eyes should be ignored as irrelevant in the description if they do not contribute to the complexity drop, whereas the missing eyebrow can be unambiguously selected as relevant.

The complexity drop principle has much explanatory power. For instance, it explains why events are more relevant if they happen in the vicinity. The reason is that distant locations are more complex to describe. It even provides a quantitative law, as complexity increases as $2 \times \log(d)$, where $d$ is the distance (see [www.simplicitytheory.science](www.simplicitytheory.science) for details). The same principle explains why events occurring close to famous landmarks (e.g., a fire at a famous cathedral) are relevant, just because the landmark makes the description of the event's location simpler.

### 3.3. Using Complexity Drop as a Principle in Machine Intelligence

From what has just been said about the importance of algorithmic information, one may conclude that machine intelligence should use Kolmogorov complexity as an ultimate guide to perform efficiently in various tasks. Data-driven learning methods such as deep-learning could be augmented beyond statistical compression. Recommendation systems could compute relevance as complexity drop, instead of merely compute frequency-based proximity. For instance, a film may be relevant to recommend based on the lower description complexity of the film for certain persons (e.g., if their name is the same as the heroine's name), independently from their proximity to previous viewers of the film. As far as compression and complexity drop can be measured, one could foresee a future for AI in which loss functions are replaced by minimum complexity computations. Transfer learning offers an illustration of this possibility. The problem consists in transferring what has been learned in a standard situation to a novel situation in which the distribution of data has changed (e.g., an autonomous vehicle in which the sensitivity of sensors is suddenly affected). Transferring learned behavior can be performed instantaneously by following a principle of minimum complexity [29]. This approach to transfer is general enough to be applied to distribution change in deep learning as well as to formal analogies.

Though the use of principles such as *minimum description length* or *minimum message length* have long been used in AI, the general use of Kolmogorov complexity in AI is hindered by the difficulty of approaching its value. The fact that Kolmogorov complexity is not computable should not, however, be regarded as a blocking factor. For instance, probability is often not computable either, as its

theoretical definition refers to sets (set of cars, set of people having a disease) that are ill-defined. And yet, various methods are commonly used to assess probability, e.g., through statistics. The same holds for complexity, which can be approached through various proxies. For instance, the complexity of an object (e.g., a famous cathedral) can be estimated by the logarithm of its rank in a list of Web search engine query outputs ordered by number of hits. Moreover, intelligent processing is more concerned with compression and complexity drop than with absolute complexity values. All one needs is to measure how many bits one can spare on a description. Such relative measures are often easy to perform and are sufficient to provide solutions. For instance, one can use complexity assessments to decide that `ppqqss` is a better answer to the analogy problem "`abc` is to `abd` as `ppqqrr` is to *X*" than other alternatives such as `ppqqrs`, `ppdqrr` or `abd` [44].

Though the search for simplicity is arguably a legitimate guide on which to base intelligent processing, a problem might arise from the fact that there is no absolute definition of complexity for finite objects. Kolmogorov complexity is defined as the size of a minimal program that a reference machine can use to compute a given object. If we change the reference machine (or, equivalently, the programming language used), one must add the length of a conversion program that translates programs from one machine to programs that can run on the other. This trick makes Kolmogorov complexity objective for objects, such as $\pi$, that present themselves as infinite mathematical series. One can say that $\pi$ has a finite complexity on any machine. For finite objects, however, the choice of the reference machine is critical, as the size of the translation program might exceed by far the complexity of the object on certain machines. In the above analogy example, it could be that `ppqqss` is the best answer for humans or for machine that implement cognitive models, but not for an autonomous machine learning system. For instance, deep learning systems used in image recognition develop primitives that are not necessarily related to ours, as suggested by the fact that they may make errors that are incomprehensible to a human eye [45].

This machine-dependence is not a real problem for AI, as the relevant machine generally comes with the problem. For instance, implementing the minimum complexity principle would be quite different if we deal with DNA sequences instead of data of human interest. There is no reason to think that our own simplicity bias, which is in part based on the recognition of algebraic group structures [46], would be able to detect DNA sequences that make sense from a functional point of view. However, trying to give our own simplicity bias to machines makes sense whenever those machines have to generate outputs that are intelligible to human beings. For instance, convolutional networks have been developed to match a human bias, just because our own vision system is insensitive to translations. Although our sense of simplicity is by no means absolute, it would certainly be a good strategy to bias machines in the same direction, if we want them to exhibit behavior that we, humans, would regard not only as efficient but also as intelligible and intelligent.

## 4. Contrast: A Missing Mechanism in Current AI Devices

A second mechanism that could improve current AI algorithms is *contrast*. Though the implementation of this mechanism is ongoing research, its potential value as a way to bridge the gap between neural network techniques and symbolic processing makes it worth mentioning.

Anyone seeing a face with a missing eyebrow or with a beard shaved only on one side cannot help but notice the weirdness of the face. Moreover, the point is not just that our attention is drawn to that face, as if it were globally salient. We know exactly why our attention has been caught and we can name the reason, the missing eyebrow or the half-shaven beard. Standard deep networks are unable of this feat. We can explain this difference by making a quite natural hypothesis: that the human brain performs a difference, a *contrast*, between what it is expecting (a symmetrical face) and what it sees, and that the missing part pops up from the contrast.

Vector difference has been used for instance in word embedding space to create insightful analogies [19]. However, mere vector difference does not always yield reliable results in this case [47]. Contrast departs from mere vector difference. Vector difference is holistic, which means that it involves

all dimensions. In holistic difference, many small discrepancies may hide a significant departure along a few dimensions, as in our example of the erased eyebrow [48]. Contrast avoids this problem by neglecting all small values (expressed in standard deviations) [49]. Thanks to this "cleaning" operation, significant differences along a few dimensions emerge from the contrast, though they are invisible in a holistic difference.

One crucial point about the contrast operation is that its output belongs to the same input space as the data. A contrast between two images is still an image. As such, contrasts may be learned and categorized as if there were original data, ideally using the same device. So if we allow a neural network to contrast an actual image with the closest typical image, the difference that emerges from the contrast can be recognized as an image in its own right. This operation would allow a network to "see" and then to recognize the missing eyebrow in a face from which it has been erased.

This new ability could prove essential to overcome several of the limitations listed above that still limit the power of deep learning. An immediate application is *anomaly* detection, as for the missing eyebrow or the half-shaved beard. It is not the only one.

The contrast mechanism was initially defined as a solution to the *inconsistency* problem [5,50]. How can we bring an image recognition system to detect a problem when seeing a house that has no door? If the missing door pops out from the contrast operation, then an essential part of the problem is solved, as some logical or causal reasoning device may then take over. Contrast is also unavoidable when it comes to *negation*. Thanks to the cleaning operation, the output of contrast is expected to be a low-dimensionality vector, if we keep only non-zero coordinates. This holds as soon as the contrasted object is close enough to its prototype. Negation, in this context, is a mechanism that performs a topological separation along this low-dimensionality direction. It may work not only for clear-cut separations, along the direction pointing to "door" in the missing door example, but also for converting gradual judgments into negation, as when an animal is judged too big to be a cat by being separated from the "cat" zone along the size axis.

These examples illustrate the fact that neural networks + contrast would ideally be able to generate *explanations*: "this is not a house, it is lacking a door", or "this is not a cat, it is too big to be a cat". Contrary to situations of binary decisions [23], these explanations could be generated in unanticipated contexts.

Contrast will be an essential tool to establish a link between statistical learning and symbolic processing. Thanks to contrast, we can generate *predicates* on the fly [50]. For instance, the predicate 'small' can be used, after applying contrast with the adequate prototype, to qualify both bacteria and galaxies. This dynamic use of predicates, thanks to contrast, solves a classical difficulty with fixed predicates, which is that the set of small galaxies is not the intersection of the set of galaxies with the set of small objects [51]. This example illustrates the fact that contrast is involved in the *semantics* of words: the meaning of 'small' relies on the contrast operation.

More generally, contrast may be crucial for learning *relations*. Relations such as `smaller(`$m$`,`$n$`)` and `larger(`$m$`,`$n$`)` could be abstracted from successive contrasts between numbers. The *systematic* opposition between these relations could be abstracted from a further contrast between (`$m$`,`$n$`) and (`$n$`,`$m$`) situations. The contrast mechanism may also be involved in a wide range of reasoning operations. It is certainly involved when processing our analogy example: "abc is to abd as `ppqqrr` is to *X*". Finding out the operation "increment last item" is easier after contrasting abc with abd.

## 5. From Reflex to Reflection

The recent AI revolution results from the fact that deep learning and other techniques such reinforcement learning, word embedding and intelligent text mining, can process real-world data. They have shortcomings, though, as the ones listed above, and these shortcomings call for qualitative improvements. In the past, much of AI efforts opposed statistical machine learning, especially neural networks, with structural matching and symbol manipulation. There is another way of conceiving the future of AI, which is to search for *mechanisms* that would operate on the fly. Several candidates have

been proposed [6,52]. I chose to mention two of them that, I think, will sooner or later be part of the solution: complexity drop, and contrast.

We are just at the beginning of exploring the potential of these techniques when they are used together. For instance, they could bring neural networks closer to *systematicity*. As we mentioned, contrast can be useful to abstract a relation such as `smaller(m,n)`, but simplicity is necessary to perform systematic generalization, by imposing that the relation correspond to a "positive" contrast, regardless of possible exceptions due to noise or errors.

The fact that neural networks are bound to perform statistical compression prevents them from the possibility of performing other forms of compression which are essential to intelligent processing. Statistical techniques rely exclusively on pre-digested expertise and function by *reflex*. By contrast, human intelligence makes use of additional mechanisms that operate on the fly, and that bring cognition closer to *reflection*. My suggestion is to make attempts to augment current machine learning techniques by implementing on top of them mechanisms that would operate dynamically (i.e., when processing new data). Coupling these mechanisms with statistical machine learning is not obvious and is the topic of ongoing research. The main point of the paper is to suggest that the mere extrapolation of current ML techniques using more computer power, more memory and more data is unlikely to produce what we would reasonably call 'intelligence', and that qualitative improvement through the introduction of dynamic mechanisms is necessary.

**Funding:** This research received no external funding.

**Conflicts of Interest:** The author declares no conflict of interest.

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
