# Peer review of "From Reflex to Reflection: Two Tricks AI Could Learn from Us"

_philosophies, doi:10.3390/philosophies4020027_

Round 1

Reviewer 1 Report

This manuscript describes several aspects of the limitations of the current deal learning methods. The author thinks that these limitations, being compared with human cognitive functions, are due to the fact that the current deep learning framework solely renders statistical compression. The points the author made in this manuscript can be helpful for researchers in the field of machine learning, although some of the claims by this author are debatable. For example, in line 138-139, the author thinks that the ability to detect structured patterns, which lies at the core of human intelligence, is beyond the reach of neural networks. However, recent works on recurrent neural networks have shown some success from the perspective of chaotic attractors. But it is also worthwhile to mention that these new works do not greatly harm the main point of this paper. Those new works are indeed beyond the statistical machine learning frameworks.

In the conclusion section, the author proposed several ingredients that may be added to the current AI research. It would be great if the author can elaborate more on how these may be realized in some concrete examples. 

Author Response

Thanks for the remarks. The paper has been significantly augmented and, I hope, improved.

I now acknowledge the fact that neural networks can indeed detect structures, and I added a recent reference about it.

I structured the paper so as to meet the last requirement. Now, complexity drop and contrast are presented as methods that may improve current AI techniques.
They are presented in much more detail than previously.

Reviewer 2 Report

This manuscript discusses limitations of current AI techniques, in particular deep learning. The manuscript argues that current techniques will not achieve dramatic breakthroughs as envisioned in concepts of “singularity”, in particular because they involve statistical reasoning that performs poorly in novel contexts. The manuscript then proposes some ideas for new types of AI techniques inspired by human cognition. The manuscript is interesting and well-written, but it lacks depth and does not explain its original contribution.

Specific comments:

One big concern with the manuscript is that it is not clear what its original contribution is. There have been many critiques of AI, from early writings of Hubert Dreyfus to recent work by Harry Collins. This manuscript’s specific critique about deep learning can already be found in the field of artificial general intelligence, which generally posits that deep learning is at most one component of a more human-like (or not-human-like) general-purpose AI. See for example papers in the Journal of Artificial General Intelligence, or works by Ben Goertzel, etc. The manuscript needs to consider this prior work and explain its original contribution; otherwise it is just rehashing old ideas and not making any progress.

Furthermore, the manuscript cites a recent book published by the author. The manuscript needs to explain the relation between that book and the manuscript. Is the manuscript a summary of ideas in the book, perhaps intended to share the ideas of the book (which is written in French) with Anglophone readers? Such a manuscript would make a different type of contribution, which may be publishable at the editor’s discretion.

A related problem is that footnote 1 cites three celebrities who are not AI experts. This is inappropriate for academic scholarship. Scholarship should not privilege the ideas of celebrities over those of experts. This is especially important because the idea in question (“the power of artificial intelligence is potentially unbounded and will probably lead to the technological singularity”) is very controversial among actual AI experts. If the author wishes to critique celebrity thinking on AI, she/he should do so in the popular media, not in the academic literature. For an academic contribution, the manuscript should respond to the views of experts in the field, which are, as is to be expected, substantially more nuanced than those of the celebrities.

Another big concern is that its discussion of new AI techniques is extremely short, just about one page of the manuscript. This is especially a concern because the manuscript title implies that the manuscript is about the new techniques. Instead, most of the manuscript is a critique of deep learning that is (to my eyes) not particularly original. It must be understood that suggesting new fundamental AI techniques is quite a big deal, and quick ideas casually tossed out like this, especially when weakly grounded in existing literature, are not reliable. The manuscript needs to unpack its ideas, relate them to existing ideas for extending AI techniques (of which there are many), and ideally also discuss their feasibility.

Some smaller comments:

“Criminal investigators would take poor decisions if” and “statistical machine learning may fail to take appropriate decisions” – “take” should be “make”

“Since such systems are blind to structure, they might propose 4,4,3,4 as a continuation” – Why 4,4,3,4? (Perhaps there is a pattern here that I am not seeing?)

Author Response

Thanks for the remarks. The paper has been significantly augmented and, I hope, improved.

Concerning the fact that several similar critiques of current AI have been published:
I now acknowledge the fact. I also added references when presenting the various arguments. I also mention that some of my points are original or are presented in an original way (what is original can be inferred from citations). Moreover, the rest of the paper (which is now significantly longer) would not make sense in the absence of the list of critiques (and I make this point clear).

I dropped the explicit reference to the book in French. Much content presented in the current version of the paper is not in the book.

I dropped the reference to celebrities when referring to the technological singularity.

Lastly, as suggested, I significantly expanded the presentation of the two techniques that are proposed to improve current AI techniques.

Round 2

Reviewer 2 Report

The revisions address my concerns and I now recommend publication pending successful response to any other reviewers’ comments.